# Is There a Limit to Growth? Comparing the Environmental Cost of an Airport's Operations with Its Economic Benefit

**Cherie Lu**

Department of Aviation and Maritime Management, Chang Jung Christian University, Tainan City 71101, Taiwan; cherie@mail.cjcu.edu.tw

**Abstract:** With the growing global awareness of the requirement for sustainable development, economic development is no longer the sole objective of business activities. The need to find a balance between environmental impacts and economic benefits is especially the case for airport operations in or around cities. This study measured the environmental costs and economic benefits and of an airport for a period of 10 years, using Taipei Songshan Airport for the empirical analysis, to examine whether the environmental costs could outweigh the economic benefits. Of all the environmental negative side effects, aircraft engine emissions and noise nuisance are considered the main sources of environmental impacts. The dose-response method and the hedonic price method, respectively, were used for estimating the social costs of these. Income generation from both direct and secondary employment is measured as economic benefits by applying the Garin-Lowry model, originally developed in 1966, for estimation of the employment multiplier. The results show that, in general, the operation of Taipei Songshan Airport brought more economic benefits than environmental costs. The sensitivity analysis of emissions and noise social cost parameters shows that the environmental costs might have exceeded the economic benefits in 2008 and 2009 in certain high emissions and noise social cost cases.

**Keywords:** airport operation; environmental impact; economic benefit; noise social cost; emissions social cost

**JEL Classification:** L93; R11

## 1. Introduction

In recent years, many national and local governments have put effort into the development of airport infrastructure, through either building new airports or expanding existing ones. This endeavor is expected to generate employment opportunities and income and to increase mobility, as well as to generate more business opportunities. Existing research has indicated that an airport generates approximately one thousand direct jobs per million passengers depending on the airport size (Airports Council International Europe-ACI Europe 2016; Airports Council International Europe-ACI Europe 2009), with variations from airport to airport. In addition, the benefits of airport operation also includes secondary effects, or the aggregation of indirect and induced economic impacts, which are the employment and income generated in the chain of suppliers of goods and services, and by the spending of incomes by the direct and indirect employees on local goods and services, respectively (Graham 2013).

On the other hand, environmental issues have also gained substantial public awareness, especially when it comes to airport operations. The negative impacts, such as noise nuisance, local pollution, greenhouse gas generation, local traffic congestion, change in biodiversity, etc.,



are unavoidable, though manageable, as long as there are flight operations (Airports Council International-ACI 2006; Airports Council International Europe-ACI Europe 2004; Postorino and Mantecchini 2016; Voltes-Dorta and Martín 2016). In the context of sustainability, however, an airport can only exist if the social and economic benefits to the region or nation that it generates are greater than its damage to human beings and the environment. Before the industry can actually decouple economic growth from environmental impacts, finding a balance between economic and socio-environmental impacts is a vital task facing all airport operators and responsible authorities.

For an airport that is right in the center of a city, its existence and positioning has always been a discussion issue for the public, as in the case of Taipei International airport, generally referred to as Taipei Songshan Airport (TSA) (Hsieh 2004). Many have argued that the valuable land in the middle of the city could be put to a better use, such as converting it to more commercial and recreational functions, than airport operation. Especially with the traffic growth in recent years, the airport has received more noise complaints than any other airport in Taiwan. In 2012, in order to control the environmental impacts of the airport effectively, TSA was the first airport in Taiwan to establish an Environmental Management System (EMS), covering issues in aircraft noise, emissions, energy onsumption, water resources, waste, ecology, ground transport, and local protection. The vision was to become a friendly and convenient capital business airport with sustainable development.

Before implementing an extreme solution such as ceasing an airport's operation and converting the valuable land it occupies to a different economic use (shopping mall, office park, etc.), one might want to examine the net social welfare that the airport brings to society. Based on the quantitative evaluation methods available, this research aims to investigate and evaluate the potential economic benefits brought by airport operations, comparing this with their environmental impact, namely those due to airport emissions and aircraft noise. Note that there are some other issues linked to airport operation, such as mobility, other environmental impacts, social equality, etc. are not considered in this research due to the uncertainty of quantifying them. In addition, the paper aims to evaluate the gross economic benefits and environmental costs of an airport. The opportunity costs of employment or airport operation are not considered in this research. For example, the staff employed by the airport could be employed in another industry or even different regions of Taiwan, or could change jobs from lower salary to higher salary, whereas the different usage of land might generate different environmental impacts. All these considerations could lead to various iterative issues. As the primary purpose of this paper is understanding the airport operation on its own, the scope of the paper is defined to evaluate the airport operation as a closed loop effect.

Both the environmental impacts and economic benefits are evaluated in monetary terms based on various mathematical models and previous research results. Regarding environmental impacts, the main focus will be on aircraft noise, aircraft engine emissions, and airport $CO_2$ emission from ground operations. The economic benefits will include employment and income generation from direct and secondary effects. TSA is used for the empirical and sensitivity analysis, with data for the 10-year timeframe from 2006 to 2015, followed by conclusions. The methodologies used in the current paper are similar to those of the paper published in 2011 by Lu (Lu 2011), which evaluated the employment benefits as well as aircraft noise and emission impacts of an airport for one year. However, the present paper has advanced on the former work by exploring the analysis over a 10-year period and applying this to a different airport. The purpose is to investigate the trends in economic benefits and environmental costs, and which circumstances might cause the environmental costs to outweigh the economic benefits that an airport brings. In addition, the $CO_2$ emissions from the ground operations at the airport have been added to the estimation of the emission social costs. All the parameters used in the empirical analysis have been carefully examined and revised according to research to date.

## 2. Methodology

### 2.1. Estimating the Social Cost of Airport Emissions

The amount of aircraft engine emissions per flight varies by aircraft operation, engine type, emission rate, flying and cruise time, and even the level of airport congestion. Exhaust emissions at airport ground level resulting from the landing and take-off (LTO) phase of flights are generally separated from the cruise level impact (European Organisation for the Safety of Air Navigation-EUROCONTROL 2005; Pagoni and Psaraki-Kalouptsidi 2016). As this research focuses on the airport level, aircraft emissions only include the aircraft operations during the LTO stages.

A number of articles in the literature (Lu 2011; Dings et al. 2005; Gallagher 2005; Schipper 2004; ExternE-Pol 2005; Dhar et al. 2009; European Organisation for the Safety of Air Navigation-EUROCONTROL 2015) have dealt with the impacts of exhaust pollutants from different aspects. With regard to their evaluation in monetary terms, they are based on the relationship between pollution and damage to human health, vegetation, buildings; climate change; and global warming. The most commonly discussed impacts are on human health and climate change. Of all the pollutants emitted from aircraft engines, six emissions—Particular Matter (PM), $SO_X$, $NO_X$, HC, CO, and $CO_2$—were found to have different degrees of negative implications for human health, with PM having the highest unit cost, and $CO_2$ the lowest. However, $CO_2$ was emitted in the greatest quantity. Therefore, the aggregated impacts of each pollutant need to take into account the unit social costs and the quantity of the pollutants during flights.

Table 1 shows a range of social costs (€/kg), listed from the lowest to the highest, for each pollutant from the selected literature. Given the uncertainty of the impacts from emissions, the evaluation reflected the wide range of monetary values. In the case of PM, the difference could be more than 100 times. The unit social cost estimates for each pollutant used in the case study below have been averaged across all the figures. However, the sensitivity analysis on the empirical results takes the range of emissions social costs into account.

**Table 1.** The unit social costs of different aircraft exhaust emissions Unit: €/kg.

| Pollutant | Average | Figures from Literature |
|:---:|:---:|:---:|
| $CO_2$ | 0.03 | 0.01, 0.02, 0.04 |
| HC | 8.86 | 0.72, 2.27, 2.70, 5.00, 8.90, 13.76 |
| CO | 0.06 | 0.01, 0.03, 0.06, 0.19 |
| $NO_X$ | 11.83 | 1.00, 1.88, 2.90, 3.20, 4.00, 11.49, 12.00, 13.00, 38.10 |
| $SO_X$ | 12.26 | 1.00, 1.53, 2.90, 3.00, 8.50, 10.73, 11.06, 40.99, 50.00 |
| PM | 292.88 | 5.09, 30.36, 75.89, 107.25, 291.81, 319.50, 600.00 |

Sources: (Lu 2011; Dings et al. 2005; Gallagher 2005; Schipper 2004; ExternE-Pol 2005; Dhar et al. 2009; European Organisation for the Safety of Air Navigation-EUROCONTROL 2015), compiled by the author. Note: the average has been converted to 2015 euros.

A bottom-up approach is applied to calculate the engine emission costs: pollutants emitted during different flight stages of LTOs are first evaluated and then aggregated for all aircraft movements for a given year (Pagoni and Psaraki-Kalouptsidi 2016). $F_{ij}$. , the kilograms of the *j*th pollutant emitted during the *i*th flight for the *k*th aircraft/engine combination is derived from

$$F_{ijk} = t_i f_{ik} e_{ijk} \tag{1}$$

where $t_i$ is the time spent during the *i*th mode of operation; $f_{ik}$ is the fuel flow during the *i*th mode for the $k^{\text{th}}$ aircraft/engine combination; $e_{ijk}$ are the emission indices of the *j*th pollutant during the *i*th

mode (kg pollutant/kg fuel) for the *k*th aircraft/engine combination. Equation (2) shows the estimation of $C_{ek}$, the social cost per flight for the *k*th aircraft/engine combination (€/flight) is

$$C_{ek} = \sum_{j=1}^{6} U_j \sum_{i=1}^{4} F_{ijk} \tag{2}$$

where $U_j$ is the per unit cost for the *j*th pollutant (€/kg). Four operational modes are calculated: taxi/idle, take-off, climb-out, and approach. Six exhaust pollutants are included (as shown in Table 1).

The main inputs related to fuel consumption of different flight modes are principally taken from International Civil Aviation Organisation (ICAO) standards. The times (minutes) for each flight mode setting for engine testing are take-off (0.7 min), climb out (2.2 min), approach (4.0 min), idle (26 min). Fuel flow (kg/s) and emission index of HC, CO and NOx (g/kg fuel) for different flight modes are included in the ICAO engine emissions databank (European Aviation Safety Agency-EASA 2015), whereas, the emission index for $SO_2$, PM and $CO_2$ are based on secondary sources (Lu 2011; European Organisation for the Safety of Air Navigation-EUROCONTROL 2015).

*2.2. Estimating Aircraft Noise Social Cost*

Aircraft noise only generates negative externalities when it creates nuisance and health impacts on people, especially for those who live in the vicinity of an airport (European Organisation for the Safety of Air Navigation-EUROCONTROL 2015). The methods generally used to capture those impacts are the hedonic price method (HPM) and contingent valuation method (CVM). The former is based on revealed behavior, and the latter is based on stated behavior, each founded on very different sets of assumptions (Pearce and Markandya 1989). The hedonic price method has been more fully developed and applied in a large number of research papers and surveys (Schipper 2004; Nelson 1980; Nelson 1981; United Kingdom Department of Environment, Transport and Regions UK DETR). This method extracts the implicit prices of certain characteristics that determine property values, such as location, environmental quality and attributes of the neighborhood. On the other hand, CVM requires a survey to be conducted for the airport concerned, which would most probably consume a certain amount of time and money. Applying HPM for evaluating aircraft noise social costs demands a set of input data and parameters, which can easily be applied to one or more airports over several years using time series data, if the data are available (Lu and Morrell 2006). As this research investigates the noise impacts for a 10-year time period, HPM is more suitable to be applied for estimating the aircraft noise social costs in monetary terms.

Applying the hedonic price method, the annual aircraft noise social cost $C_n$ can be derived as

$$C_n = \sum_i I_{NDI} P_v H_i (N_{ai} - N_0) \tag{3}$$

where $I_{NDI}$ is the noise depreciation index (NDI) expressed as a percentage per dB(A)[1]; $P_v$ is the annual average house rent within the noise contour; and hence, $I_{NDI} P_v$ is the annual aircraft noise cost per residence per dB(A). This is multiplied by $H_i$, the number of residences within the *i*th zone of the noise contour. $N_0$ is the background noise or the ambient noise, where $N_{ai}$ is the average noise for the *i*th section of the noise contour; the noise level above the ambient level is then $(N_{ai} - N_0)$ (Lu and Morrell 2006). One of the main inputs used here for evaluating the cost of noise at an airport is the NDI, which is the percentage reduction of house price per dB(A) above the background noise, 50dB applied for this research. Based on the literature review results (Hsieh 2004), the average NDI is assumed to be 0.6%.

---

[1]  dB(A): the A-weighted decibel units, adjusted to conform with the frequency response of the human ear.

### 2.3. Methods for Evaluating Income Generation of Airport Operations

The models most often used for assessing the economic benefits generated in a specific geographical area by a given industry are the input-output model (the I-O model) and the Garin-Lowry model. The Garin-Lowry model of 1964 was one of the first transportation/land-use models to be developed (Garin 1966). The gravity-based Garin-Lowry model dealt with urban population and employment distribution for a given pattern of basic employment, with some revisions and extensions later on by other researchers (Guldmann and Wang 1998; Jun and Moore 2002). The I-O model is generally applied to predict the effect on the economy of changes in one industry (Airports Council International Europe-ACI Europe 2016; Lee 2005). Considering the purpose of this research and data availability, the Garin-Lowry Model was found to be suitable in estimating the employment generation of an airport.

Employment is assumed to come from either "basic" or "non-basic" industry. "Basic" industry is the administrative, industrial, and business activities that do not depend on a local market for their employment. "Non-basic" industry covers those activities such as hospitals, schools and shops whose employment is a function of the local economy (Pagoni and Psaraki-Kalouptsidi 2016). The underlying assumption for the economic-based multiplier approach is that employment in the basic sector will produce increases in a region's population, which will in turn increase employment in the non-basic sector. The total employment ($E$) generated by the operation of an airport is then the sum of the basic (or direct) employment ($E_B$) and the non-basic (or secondary) employment ($E_{NB}$). The base multiplier ($M$) is the measure of overall employment as a percentage of employment in the basic industry. A generalized form is expressed as in the Equations (4) and (5),

$$E = E_B + E_{NB} \tag{4}$$

$$M = E/E_B \tag{5}$$

Furthermore, two geographical areas, called the "reference area" and the "region" within the reference area, are considered (Pagoni and Psaraki-Kalouptsidi 2016). In the case of this study, the region is Taipei City, where TSA is located, and the reference area is the whole of Taiwan. The simplified Garin-Lowry model expresses the total employment due to the basic industry as given in Function (6) (Lu and Liu 2014; Garin 1966).

$$E = E_B + E_{NB} = E_B + \frac{\alpha \beta E_B}{1 - \alpha \beta} = E_B (1 - \alpha \beta)^{-1} \tag{6}$$

where,

$\alpha$: Employment-dependency ratio, $\alpha = N/E_R$, calculated by using the regional population, $N$, divided by total employment in the region ($E_R$);

$\beta$: Non-basic employment ratio; $\beta = E_a/E_R$, calculated by taking the total employment in the non-basic industry ($E_a$) divided by total employment in the region. (Note that the service industry is later found to be the basic industry for Taipei in this case—see Section 3.3).

The value of $\beta$ is dependent on the results of the Location Quotient approach, which interprets the characteristics of a specific industry and assesses the intensity of specialization of the local economy from employment data of the region and the reference area (Pagoni and Psaraki-Kalouptsidi 2016; Lu and Liu 2014). It is expressed in Equation (7),

$$LQ_i = \frac{e_{iR}/E_R}{e_{iT}/E_T} \tag{7}$$

　　　$LQ_i$ : *Location Quotient*
　　　$e_{iR}$ : *industry i's employment in the region*
where　$E_R$ : *Total employment in the region*
　　　$e_{iT}$ : *industry i's employment in the reference area*
　　　$E_T$ : *Total employment in the reference area*

Depending on the value of $LQ_i$, an industry could be classified as either non-basic or basic:

- $LQ_i$ = 0, shows that there is no industry *i* in this region;
- $LQ_i$ > 1, indicates that industry *i* has a tendency to be concentrated and specialized; a basic industry;
- $LQ_i$ < 1, shows that industry *i* has no concentrated and specialized trend; a non-basic industry.

The smaller the $LQ_i$ value, the less important the *i*th industry is. The larger $LQ_i$ is, the more the *i*-th industry is specialized and centralized, and hence more important.

The defining economic benefits of an airport are the employment and income generation. With the derived employment figures, the income generation is the sum of direct employment and secondary employment, each multiplied by the average salaries of air transport-related and other industry employment respectively, as show in Equation (8):

$$E_E = (E_B \times S_1) + [(E - E_B) \times S_2] \qquad (8)$$

where, $E_E$ represent airport economic benefits; $S_1$ is the average salary of air transport employment; $E - E_B$ means secondary employment; and $S_2$ is the average salary of other industry employment.

## 3. Empirical Analysis—Taipei Songshan Airport

TSA is owned and managed by the Taiwan Civil Aeronautics Administration (CAA). In 2015, it handled 5.8 million passengers and 57,597 aircraft movements. The historical traffic volumes are shown in Table 2, with the highest passenger volume of more than 6 million in 2006. The sharp drop in the consequent years was mainly due to the operation of Taiwan High Speed Rail from Taipei to Kaohsiung, which has quickly captured all the North-South travel of the island's west corridor. However, with the liberalization of cross-strait flights and short intra-Asian routes, the traffic level quickly increased in 2011 and has been stable in the past several years.

**Table 2.** Traffic volumes at Taipei Songshan Airport from 2006 to 2015.

| Year | Passenger Traffic | Aircraft Movement | Cargo (Tonnes) |
|------|-------------------|-------------------|----------------|
| 2015 | 5,861,902 | 57,597 | 45,217 |
| 2014 | 6,105,403 | 61,881 | 43,528 |
| 2013 | 5,847,275 | 60,066 | 36,319 |
| 2012 | 5,676,411 | 58,170 | 31,311 |
| 2011 | 5,258,975 | 58,185 | 34,492 |
| 2010 | 3,712,841 | 48,925 | 14,355 |
| 2009 | 3,091,066 | 44,655 | 11,405 |
| 2008 | 3,101,854 | 49,264 | 11,830 |
| 2007 | 4,470,859 | 68,084 | 13,115 |
| 2006 | 6,728,709 | 87,955 | 15,024 |

Source: Taiwan Civil Aviation Statistics (2016).

### 3.1. Airport Emission Costs

The aircraft operated from TSA are generally narrow-bodied aircraft (i.e., B737, A320, MD90, etc.), turbo-props (i.e., ATR-72), regional jets (i.e., ERJ-190), and business jets. However, with the strong demand of short-haul city-pairs, there have been some operations of wide-bodied aircraft, namely A330s. Applying the mathematical model in Section 2.1, aircraft engine emission costs from the most

commonly used aircraft types are listed in Table 3. The engines used for each aircraft type are taken from those airlines operating out of TSA. The costs are LTO stages, using ICAO aircraft standard operating procedures. The possibility of different aircraft ages resulting in different emission levels is not considered in the analysis, as there is no publicly available data in this regard. The average emission cost for the aircraft types at TSA ranges from €872 of a B787 landing and take-off, to an average business jet at €167. Multiplying the engine emission costs of different aircraft types by the number of aircraft LTOs for each type provides an overall annual social cost from aircraft engine emissions.

**Table 3.** Average emission cost for commonly used aircraft types at Taipei Songshan Airport.

| Aircraft Type | Cost per LTO (Euros/LTO) | Aircraft Type | Cost per LTO (Euros/LTO) |
|---|---|---|---|
| B737 | 296 | DH8 | 225 |
| B757/767 | 583 | ATR72 | 108 |
| B787 | 872 | ERJ-190 | 177 |
| A320 | 254 | MD-90 | 315 |
| A321 | 375 | MD-82/83 | 311 |
| A330 | 756 | Business jets | 167 |

Note: Where different engines are used by airlines, the actual engine emission indices are applied wherever possible. The figures for some aircraft types (MD-82/83, A320, B737, A330, business jets) in this table are given as averages.

Furthermore, according to the airport emission inventory data from Taiwanese airports of similar size to TSA, the aircraft $CO_2$ emissions from LTO procedures account for half of the total airport $CO_2$ emissions (Ju 2016). Therefore, assuming the rest of the airport $CO_2$ emissions are the same as aircraft LTO $CO_2$ emissions, the total airport emission cost is aggregated as in Table 4, varying from €5.4 million in 2009 to €12.9 million in 2006.

**Table 4.** Annual average aircraft engine emissions social cost for 2006–2015.

| Year | Aircraft LTO | Euros per LTO | Aircraft Emission Cost (m Euros/Year) | Other Airport $CO_2$ Cost (m Euros/Year) | Total Emission Cost (m Euros/Year) |
|---|---|---|---|---|---|
| 2015 | 28,799 | 321 | 9.3 | 2.6 | 11.9 |
| 2014 | 30,941 | 321 | 9.9 | 2.8 | 12.8 |
| 2013 | 30,033 | 279 | 8.4 | 2.4 | 10.8 |
| 2012 | 29,085 | 278 | 8.1 | 2.3 | 10.4 |
| 2011 | 29,093 | 247 | 7.2 | 2.1 | 9.3 |
| 2010 | 24,463 | 167 | 4.1 | 1.4 | 5.5 |
| 2009 | 22,328 | 183 | 4.1 | 1.3 | 5.4 |
| 2008 | 24,632 | 199 | 4.9 | 1.5 | 6.4 |
| 2007 | 34,042 | 211 | 7.2 | 2.2 | 9.4 |
| 2006 | 43,978 | 223 | 9.8 | 3.1 | 12.9 |

Note: The average costs per LTO for 2007 and 2009 are the average of the years before and after, as no aircraft fleet mix data are available for these two years. Monetary values are given in 2015 euros.

*3.2. Aircraft Noise Social Costs*

According to the Taiwan Environmental Protection Agency (EPA) regulations, 10 Taiwanese airports (out of 17 in total) that generate significant noise nuisance have published noise contours and modified them periodically. Within the noise contour, three aircraft noise control zones are classified, and their respective criteria are

- Class 1: areas between 60-65 dB(A) of aircraft noise day-night average sound level ($L_{dn}$);
- Class 2: areas between 65-75 dB(A);
- Class 3: areas exposed to noise higher than 75 dB(A).

The number of households affected by aircraft noise within different zones in Table 5 is derived by taking the estimated number of households affected for 2012 (Lu 2014) and adjusting it by using the published population data (number of households) in the administrative areas in these zones for each

of the years. Annual house rent is estimated from the publicly available house price data for the all the administrative areas within noise zones weighted by the households, considering the average house age and discount rates. An average house age of 10 years was used, based on the housing statistics for Taipei, whereas discount rates of 2–3% were applied using the nominal interest rates in Taiwan for reference (Lu and Liu 2014). Following the noise-social-cost estimation method, substituting the average house rent and the number of residences within each noise zone, the noise costs are estimated for 2006–2015, as shown in Table 5, ranging from the lowest at €18.6 million in 2006 to the highest at €33.0 million in 2015.

**Table 5.** Noise social cost at Taipei Songshan Airport from 2006 to 2015.

| Year | Annual House Rent (Euro/Year) | Estimated Households within 60 $L_{dn}$ Contour | Noise Social Cost (m Euros/Year) |
|---|---|---|---|
| 2015 | 14,191 | 25,501 | 33.0 |
| 2014 | 14,224 | 25,423 | 33.0 |
| 2013 | 14,320 | 25,345 | 33.1 |
| 2012 | 14,092 | 25,267 | 32.5 |
| 2011 | 14,070 | 24,810 | 32.0 |
| 2010 | 12,574 | 24,473 | 28.3 |
| 2009 | 10,937 | 24,318 | 24.5 |
| 2008 | 11,718 | 24,185 | 26.1 |
| 2007 | 10,407 | 24,093 | 23.1 |
| 2006 | 8368 | 24,053 | 18.6 |

Note: Monetary values are given in 2015 euros.

*3.3. Employment Opportunities and Income Generation*

The Directorate General of Budget, Accounting and Statistics (DGBAS) and National Statistic, ROC (Taiwan), divide the industrial activity of the employed population into primary, secondary, and tertiary industries.

- Primary industries include agriculture, agribusiness, forestry, fishing, mining, and quarrying industries;
- Secondary industries are those such as aerospace and automobile manufacturing, brewing and the tobacco industry, the chemical industry, the textile industry, the energy industry, steel production and metalworking, industrial equipment production, and consumer electronics;
- Tertiary industry is the service industry.

Table 6 shows the industry employment and the subsequent location quotient for 2014, as an example. In fact, for the years 2006–2015, primary and secondary industries both have LQ values less than one, with location quotient values of 1.37–1.40 for tertiary industry. Since we are focusing on TSA, which is a service industry, as our basic industry, the tertiary industry population is used in our analysis.

**Table 6.** Employment by industry for Taipei city and Taiwan—2014.

| Industry | Taipei City Employment | | Taiwan Employment | | Taipei Location Quotient |
|---|---|---|---|---|---|
| | (000) | % | (000) | % | |
| Primary Industry | 2 | 0.2 | 548 | 4.9 | 0.03 |
| Secondary Industry | 238 | 18.8 | 4004 | 36.2 | 0.52 |
| Tertiary Industry | 1023 | 81.0 | 6526 | 58.9 | 1.38 |
| Total | 1263 | 100.0 | 11,078 | 100.0 | - |

Source: Taiwan National Statistics (2016), Taipei City Statistics (2016).

Based on the historical employment data given by TSA, the average employment per million passengers is increasing over the years, from 450 to 900, covering airports, airlines and ground handling agents as well as companies linked to the airport. This could be explained by the commercialization of airport operations as well as more Taipei Mass Rapid Transit lines connecting through the airport.



For comparison, recent employment is just slightly less than the average employment value of 950 per million passengers for European Airports of similar size (Airports Council International Europe-ACI Europe 2016).

The total employment can then be evaluated by applying the Garin-Lowry model in Equation (6). Using year 2014 as an example, full-time equivalent employment at the airport was 5190 and the population of Taipei city was 2.702 million, with 1.023 million of a total of 1.262 million employed in the tertiary industry (Table 6). These values give an $\alpha$-value of 2.14[2] and $\beta$-value[3] of 0.19 and thus total employment benefits of

$$E = E_B \left( \frac{1}{1 - \alpha\beta} \right) = 5190 \times \left( \frac{1}{1 - 2.14 \times 0.19} \right) = 8755 \tag{9}$$

Of these 8755 jobs, 3565 (total employment 8755 minus direct employment 5190) can therefore be considered as the secondary employment. The employment multiplier for TSA is then estimated to be 1.69, as shown in Equation (10), which lies within the reasonable range of American and European airports with similar traffic volumes (Graham 2013).

$$M_B = \frac{E}{E_B} = \frac{8755}{5190} = 1.69 \tag{10}$$

Over the years of 2006–2015, the $\alpha$-values slightly decrease from 2.30 in 2006 to 2.12 in 2015, which could be partially due to the decline in birth rates in Taiwan. On the other hand, the employment by different categories of industries have been stable, therefore, the $\beta$-values are considerably the same, around 0.19–0.20 over the years.

Applying the data from DGBAS, the average annual income per full-time employee in the air transport sector in Taipei in 2014 was €44,065, suggesting a direct economic benefit of €228.7 million. Based on various income statistics for neighboring regions and industries, the average annual income for secondary employment was €23,817, which represents a secondary income generation of some €84.9 million. TSA's operations in 2014 therefore provided a total economic benefit of around €313.6 million.

Table 7 lists the estimation of total employment, direct, secondary and total income benefits from the operation of TSA from 2006 to 2015. The total employment generated by the operation of the airport is estimated to be the lowest at 2952 in 2008 and the highest at 8853 in 2015. The total income benefits range from the lowest level of €102.1 million in 2008 to the highest of €320.8 million in 2015. As direct employment increases, leading to higher indirect employment, the multiplier therefore decreases over the years.

**Table 7.** The estimation of employment and income benefits from the operation of Taipei Songshan Airport.

| Year | Total Employment | Multiplier | Direct Income (m Euros/Year) | Secondary Income (m Euros/Year) | Total Income (m Euros/Year) |
|---|---|---|---|---|---|
| 2015 | 8853 | 1.68 | 234.8 | 86.0 | 320.8 |
| 2014 | 8755 | 1.69 | 228.7 | 84.9 | 313.6 |
| 2013 | 8100 | 1.73 | 202.5 | 80.1 | 282.6 |
| 2012 | 7315 | 1.72 | 185.4 | 72.0 | 257.4 |
| 2011 | 6424 | 1.75 | 165.6 | 63.6 | 229.2 |
| 2010 | 4263 | 1.77 | 101.0 | 44.2 | 145.2 |
| 2009 | 3279 | 1.77 | 76.7 | 33.7 | 110.3 |
| 2008 | 2952 | 1.73 | 72.0 | 30.1 | 102.1 |
| 2007 | 3965 | 1.77 | 95.7 | 42.3 | 138.0 |
| 2006 | 5533 | 1.83 | 112.6 | 59.6 | 172.2 |

Note: Monetary values are given in 2015 euros.

---

[2]   Taipei population divided by total employment in Taipei—see Table 6.
[3]   The employment of non-basic (primary and secondary) industries divided by total employment in Taipei—see Table 6.

### 3.4. The Comparison of Environmental Costs and Economic Benefits

Combining the results of income generation and environmental costs, both noise and airport emissions, Figure 1 illustrates the trends over the past ten years. However, it is worth noting that the benefits of an airport are widely dispersed across the economies, while the environmental costs are more concentrated geographically on the people in the vicinity of an airport. These social effects are not further explored in this paper.

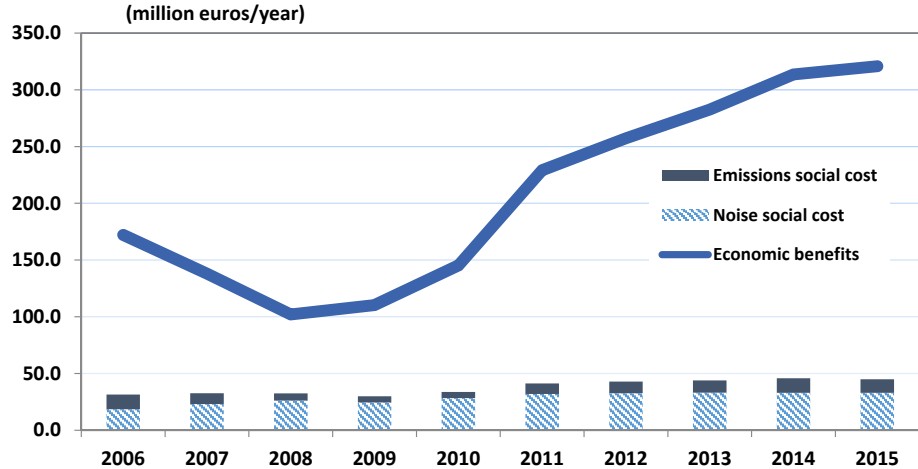

**Figure 1.** The estimation of economic benefits and social costs of Taipei Songshan Airport for 2006–2015~Base case.

The biggest gap between economic benefits and environmental costs is found in 2015, with net benefits of €276 million. Even in 2008 with the least difference, the net benefit is estimated to be €70 million. It has been increasing over the past years. The fact that 2008 had a comparatively low traffic volume brings down the economic benefit; on the other hand, house rent was relatively high in that year, which results in a high estimation of the noise social cost. Since 2011, the gap between economic benefits and environmental costs has been similar, as the key airport operation environmental and socio-economic factors have been reasonably stable.

Furthermore, considering key parameters for noise and emissions models for sensitivity analysis shows in which circumstances the environmental costs would exceed economic benefits. With regard to noise social cost, the NDI value of 0.6% is used in the base case. Research has shown that the NDI value for TSA might range between 0.58% and 2.57% (Lu and Liu 2014). A sensitivity analysis on the NDI value shows that the environmental cost would be higher than economic benefit in years 2008 and 2009 if the value were 2.0%, as shown in Figure 2. On the other hand, the literature review shows that there is a wide range values for the unit social costs for different pollutants. However, even the unit social cost for all pollutants have to be 12 times the base case for negative net economic benefits to occur (as shown in Figure 3), which might not be likely, based on current findings of the damage from these pollutants.

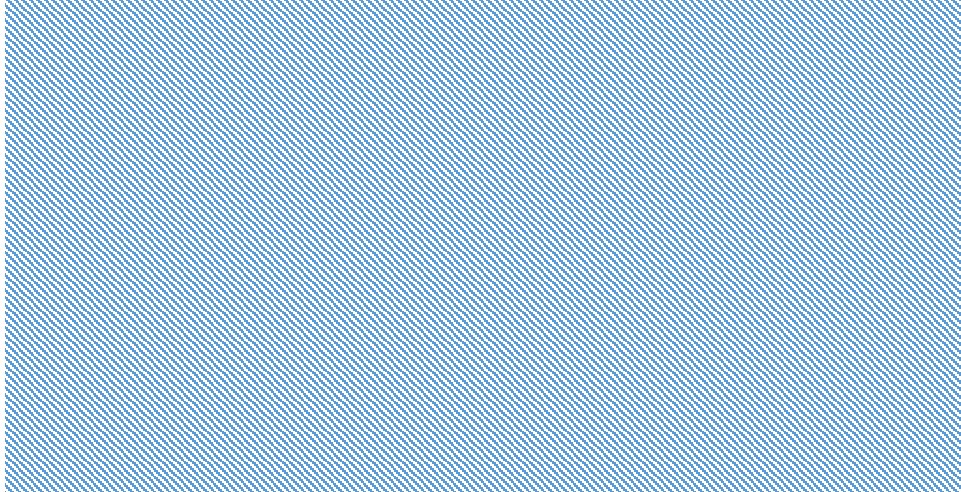

**Figure 2.** The estimation of economic benefits and social costs of Taipei Songshan Airport for 2006–2015~with NDI value of 2.0%.

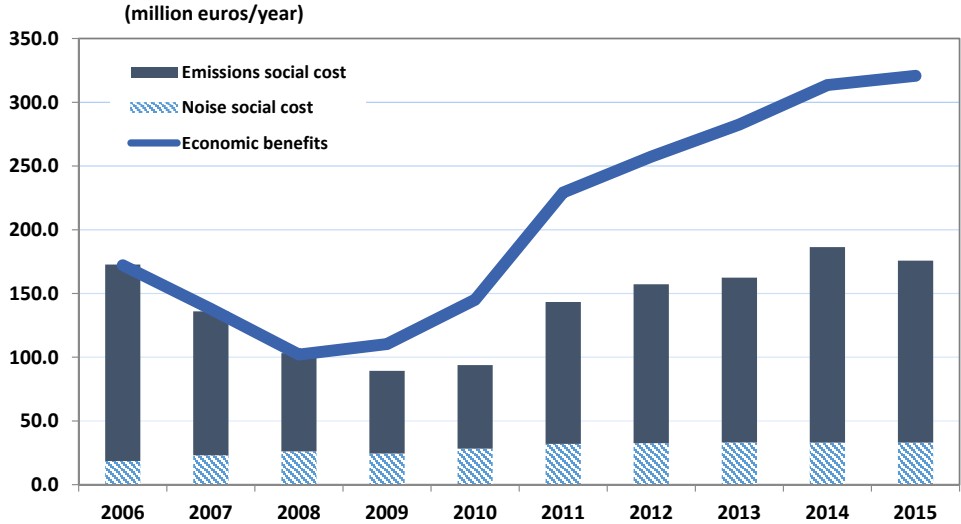

**Figure 3.** The estimation of economic benefits and social costs of Taipei Songshan Airport for 2006–2015~12 times unit social costs of all pollutants.

## 4. Conclusions

While seeking economic growth is not the sole objective of an activity, the negative impacts, such as environmental externalities, are often at the core of public debate. The existence of TSA, right in the centre of Taipei city, has generated much discussion and many different opinions over the past twenty years: from the functions of the airport, to the size of the airport, even voices about closing the airport completely for different uses. This paper has evaluated the economic benefits and environmental costs of an airport, using TSA as a case study. Note that there are some other positive or negative aspects of airport operations that are not considered in this analysis. However, the generalized research methodology applied in the paper could easily be applied at any airport, thus increasing awareness of the airport's environmental impacts and of the results of applying environmental management measures.

With regard to environmental costs, the noise social cost is on average three times more than airport emission cost. The research also shows that TSA's average net economic benefit has been around €200 million annually for the past 10 years, with 2008 having the smallest gap between economic benefits and environmental costs. This shows that the airport acts clearly as a catalyst for economic

development in the region through indirect economic effects, even during the period of declining airport traffic and economic activity in 2008 and 2009.

On average, the economic benefit is more than five times the aggregation of both airport emissions and aircraft noise social costs. The increase of traffic in recent years, although bringing more environmental impacts, still generates more economic benefits than the social costs of airport emissions and aircraft noise combined. The sensitivity analysis shows that only when there is a high NDI value and extreme emissions unit social costs would environmental costs outweigh the economic benefits. Based on the current literature regarding the unit social costs of pollutants, the emission social cost from airport operations does not seem to be the dominant factor in exceeding economic benefits, unless there are more dramatic scientific findings about the damage of those emissions, especially on climate change, human health, and ecology.

The paper attempted to measure the employment benefits of an airport and compare these with environmental costs for a period of 10 years. The results and sensitivity analysis have properly demonstrated the purpose. Some assumptions and limits on the scope are necessary in order to complete the analysis within a reasonable timeframe and resources. Further research could explore the social impacts of an airport or apply noise social costs from health impacts. As noted in the introduction, this paper did not evaluate the opportunity cost of employment, which might lead to overestimating of economic benefits. Therefore, one could also look into the opportunity costs of an airport, for example the opportunity cost of employment or the land the airport occupies.

**Acknowledgments:** The author would like to thank Jo Chen for collecting the related socio-economic data and Chih-Gang Liu for compiling the data. This research is part of a research project funded by the Taiwan Ministry of Science and Technology [MOST 104-2410-H-309-011-]. The preliminary results of this research were published in Mandarin in Maritime Quarterly in June 2014.

**Conflicts of Interest:** The author declares no conflict of interest.

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
