# Peer review of "Is There a Limit to Growth? Comparing the Environmental Cost of an Airport’s Operations with Its Economic Benefit"

_economies, doi:10.3390/economies5040044_

Round 1

Reviewer 1 Report

Abstract: Abstract provides a concise overview of the paper, with sufficient detail to inform the audience and compel further reading. Year required for Garin-Lowry model (P1, L16). Minor grammatical/spelling error noted (ie. P1, L11).

Introduction: The assertion that airports at the centre of cities are discussion issues for the public (P2 L46-47) creates an assumption that airports removed from urban context are less prone to attracting discussion. This is not the case (i.e. Amsterdam Schiphol is one of the most highly deliberated airports in the world, despite its physical separation from the city). Consider reframing the argument around airports as normatively attracting public debate for NIMBY or social/environmental impacts.

The introduction notes “evaluation methods available”(P2 L55-56) should be changed to “quantitative evaluation methods available” as this will enable the author(s) to remove the misunderstanding that social and environmental impacts can only be robustly evaluated via quantitative methods (as is implied in P2 L58-60).

Methodology: Paragraph spacing and font size is inconsistent in Pages 3, 4 and 5 (between formulae).

Empirical Analysis: Please identify TSA as a “government owned and managed airport”, rather than a “traditional airport” (P5 L195). The source for the assertion on P6 L220-221 should be formally referenced in-text. 

Consider reducing the number of figures, as when they are presented one after the other as they are currently, it creates some difficulty in cognitively navigating the analysis.

The data suggests that even during periods of declining economic activity (i.e. 2008-2009), TSA provides a positive economic benefit for the city. This phenomenon needs to be explained in some detail, as there is a point of discussion here that is under-sold for its contribution to the literature. That is, the data clearly indicates that the airport acts as a catalyst for economic development/benefit through the indirect economic effects that aviation has on the city’s industries (i.e. the growing gap between social costs and economic benefits over the course of the data set analysed – Figure 3 is the clearest on supporting this point). Please consider giving a few more sentences to elaborate this in the conclusions section to provide further direction for future research.

Conclusions: Concise and accurate to the scope of the study, although as pointed out in previous comments, there is some more benefit to the paper than the author(s) elaborate in the conclusion.

Author Response

Reviewer 1:
Comments and Suggestions for Authors
Abstract: Abstract provides a concise overview of the paper, with sufficient detail to inform the
audience and compel further reading. Year required for Garin-Lowry model (P1, L16). Minor
grammatical/spelling error noted (ie. P1, L11).
Answer:Thank you for your valuable comments. Have added the year for the
Garin-Lowry model and corrected the spelling error.
Introduction: The assertion that airports at the centre of cities are discussion issues for the
public (P2 L46-47) creates an assumption that airports removed from urban context are less
prone to attracting discussion. This is not the case (i.e. Amsterdam Schiphol is one of the most
highly deliberated airports in the world, despite its physical separation from the city). Consider
reframing the argument around airports as normatively attracting public debate for NIMBY or
social/environmental impacts.
Answer: In order to avoid confusion, the revised paper has added several sentences
to the situation of Songshan Airport in the same paragraph.
The introduction notes “evaluation methods available”(P2 L55-56) should be changed to
“quantitative evaluation methods available” as this will enable the author(s) to remove the
misunderstanding that social and environmental impacts can only be robustly evaluated via
quantitative methods (as is implied in P2 L58-60).
Answer:Thank you for the valuable comment. The word “quantitative” has been
added to the sentence.
Methodology: Paragraph spacing and font size is inconsistent in Pages 3, 4 and 5 (between
formulae).
Answer: Because the formulas, the spacing and font size are different. It is hoped
that the editorial office can standardise it.
Empirical Analysis: Please identify TSA as a “government owned and managed airport”, rather
than a “traditional airport” (P5 L195). The source for the assertion on P6 L220-221 should be
formally referenced in-text.
Answer: Have revised the sentence and added the reference (which was through
personal communication).
Consider reducing the number of figures, as when they are presented one after the other as
they are currently, it creates some difficulty in cognitively navigating the analysis.
Answer:Apart from the base case in Figure 1, Figures 2 and 3 represent the changes
in noise and emissions social costs respectively. Since it is easier to identify
the trends and changes through figures rather than tables, the results of
these sensitivity analyses are only presented in figures. The author could
not find a better way to present the results.
The data suggests that even during periods of declining economic activity (i.e. 2008-2009),
TSA provides a positive economic benefit for the city. This phenomenon needs to be explained
in some detail, as there is a point of discussion here that is under-sold for its contribution to the
literature. That is, the data clearly indicates that the airport acts as a catalyst for economic development/benefit through the indirect economic effects that aviation has on the city’s
industries (i.e. the growing gap between social costs and economic benefits over the course of
the data set analysed – Figure 3 is the clearest on supporting this point). Please consider
giving a few more sentences to elaborate this in the conclusions section to provide further
direction for future research.
Conclusions: Concise and accurate to the scope of the study, although as pointed out in
previous comments, there is some more benefit to the paper than the author(s) elaborate in
the conclusion.
Answer: The revised version has added more discussion about the results and
further research recommendations in the conclusions.

Author Response

Reviewer #2
Review of Is there a Limit to growth? Comparing the environmental cost of an
airport’s operations with its economic benefit.
This paper uses a time period for a single airport (Taipei Songshan Airport) to assess
whether over time the relative values of economic benefits and costs change over
time. Intuition might suggest that diminishing economic benefits would occur and
environmental costs may increase given the increased value placed on the
environmental parameters. This is a well written paper that does a good job in
examining what I might term the straight forward benefits and costs of an airport,
however I see show shortcomings and I have questions.
1. A key issue in environmental analysis particularly in a benefit-cost framework is
aggregation. BC aggregates over disparate measures of benefits and costs all in $
units and yields a net present value or B/C ratio or marginal B/C ratio. However,
the benefits of an airport are widely dispersed while the costs are more
concentrated. Aggregation ignores this important fact. The paper should at least
discuss this issue since it can and does affect government decisions and has
contributed to the development and application of the use of MAE, multiple
accounts analysis. This needs o tbe discussed.
Answer:Thank you for your valuable comments. Yes, this research, simply
aggregating the environmental costs and comparing against the economic
benefits, does not take into account the dispersed effects. This issue is
addressed in the first paragraph of Section 3.4.
2. In looking at emissions the author only looks at aircraft emissions yet a significant
amount of emissions does or has come from on airport vehicles and another
significant source will be front vehicles delivering people and goods to and from
the airport (ground transportation). Why are these ignored when they can and
likely do result in a significant underestimate of the social cost of emissions?
Answer: Apart from the aircraft engine emissions, this paper does take into account
the airport CO2 emissions from ground operations, as was stated at the
final paragraph of Section 1. The final paragraph of Section 3.1 discusses
the incorporation of airport CO2 emissions based on the real airport data.
However, the title of Section 3.1 has been revised to “Airport emission
costs” in order to reflect the results more suitably.

3. The author uses landing and takeoffs to measure activity levels yet newer aircraft
not have lower noise levels but also much lower emissions as they burn
significantly less fuel. Therefore simply counting number of operations will bias
the cost measure of both emissions and noise unless the proportion of newer
aircraft used are taken account of.
Answer: Yes, ideally it is better to take into account different ages of aircraft fleet for
calculating aircraft noise and engine emissions. However, there are no
publicly available data regarding the relationship between aircraft age and
noise/emission levels. Therefore, the standardized ICAO data are used. This
issue is also addressed now in the first paragraph of Section 3.1.
4. The Garin-Lowry model is a very old model and has several limitations. It is
notably a static model, which does not tell anything about the evolution of the
transportation / land use system. Furthermore, recent economic changes are in
the service (non-basic) sectors, forming the foundation of urban productivity and
dynamics in many metropolitan areas, cannot be effectively represented. Under
such circumstances the model is likely to be inaccurate in major service-oriented
metropolitan areas; is this true of Taipei? Further, the Lowry model does not
consider movements of freight in urban areas, which are very significant and
have impacts on the friction of distance. How do these limitations affect the
benefit measures? Particularly since the author is interested in the change sover
time, yet the model is holding economic structure constant.
Answer: Yes, Taipei is a service industry based city. The β (Non-basic employment
ratio) value in equation (6) takes into account change of non-basic
employment ratio, which means the changes in the service industry in
Taipei are incorporated in the model. The multiplier (equation (10))
considers the generation of employment and income from the income of
the direct employment, which implies the movement of freight and goods
associated with these activities.
5. How does the model account for aviation’s use of the airport affecting airport
employment? Again, operations is not complete since it does not consider
domestic from international operations. Each has a different impact on airport
and regional employment.
Answer: Indeed, this paper only considers the employment of the airport operation,
covering direct and indirect employment and income. Therefore, the total
employment of airport in a given year is considered. However, this covers
both domestic and international parts of the airport operation. The airport used to be a domestic airport, the cross-strait direct flights started
gradually in 2008. At the end of 2010, there started short haul intra-Asian
direct flights operation. Table 7 shows a big increase in direct income at
the airport from 2011. This would imply the increase of international
flights starting from that time.
6. I am skeptical of the noise cost calculation. Economic theory would predict that an
externality such as noise would impose an economic cost on people in several
different ways. First, property depreciation, assets such as homes would become
less valuable in the marketplace because the asset now has a lower quality of
services flowing from it. Equivalently, in order to enjoy the same level of services
such as outdoor relaxation, sleeping or conversation resources would have to be
invested. Second, the transactions cost associated with moving to another
location impose an economic loss on those who decide to move. Third, people
who decide to move will face a cost of lost utility in the form of attachment to
their home. This would be captured in a measure of lost consumer surplus. Fourth,
people who remain in the area, for whatever reason, face increased nuisance
noise and, therefore, a reduction in the flow of services from their homes. The
magnitude will depend upon a number of factors including sensitivity to noise and
activities that are perceived affected by noise such as sleep, recreation or solitude.
Below a conceptual framework is presented which provides empirical measures of
each of these components. I think it is underestimated in some respects and
over-estimate in others (due to not accounting for the proportion of new aircraft
using the airport). There are four elements to consider; in principle, the impact on
the different types of housing ) detached owner owned), semi-detached and
rental, the division between those who move out of the environment and those
who choose not to do so, the impact of noise as measured from an existing base
case developed from current runway usage and the discounting of all measured
costs over a time horizon which reflects when the traffic changed. The base case
from which all measurements are made is established and hence traffic levels and
composition. This level of airport activity will have imposed property depreciation
and noise annoyance costs upon the airport environs and it is from this level that
the 'incremental costs' of the alternative runway configurations are calculated. I
also do not understand how the author established the price of noise or cost of
quiet. This is well established in the literature. Furthermore, it is the incremental
change in trffic that affects noise not the total traffic each year since previous
traffic affects are already fully capitalized into the property value. An example
calculation would be:

please refer to pdf file (could not upload formulas in the system).

Answer:Thank you for this valuable contribution. The noise social cost calculation is
based on lots of assumptions and has taken average values instead of
separate cases. The parameters used for the noise social cost model are
based on research results of Songshan Airport, which have considered the
characteristics of the area. The results of annual noise social costs in Table
5 are based on the annual house rent, which tries to capture the
annoyance costs on the residents of that year. Since the residents have to
endure the noise every year, this research is based on the assumption that
the noise social costs are repeating every year, covering all the traffic of a
given year.
7. Figure 1 is confusing. Why would economic benefits continue to rise at such a
high rate over time. It goes against the whole idea of diminishing returns and it
seems to be implicitly assuming some sort of structural change in the economy.
Answer: Since the economic benefits are the aggregation of the income generated
by direct and indirect employment, the employment numbers at the
airport are important parameters. As shown in Table 7, the direct income
has increased sharply from 2011. This would lead to the big increase of
economic benefits since then.

Reviewer 3 Report

General comments:
This paper compares the environmental cost of airport’s operation with its economic benefit, using a case of Taipei’s airport. Analytical framework is quite straightforward. My opinion is the estimation of economic benefit is too simple to be biased. The study assumes that all newly generated employments are hired at full wage rates (or average salary); and it means that the study assumes such the newly employed labors would be unemployed if the airport would not be operated. However, the employment generated by the airport operation could be hired in another industry when the airport would not be operated. In other words, the change of jobs from less salary to higher salary seems ignored. It is highly afraid that this ignorance leads to serious overestimation of economic benefit stemming from the airport operation. Both the change from unemployment to employment and the change from less efficient job to more efficient job should be explicitly incorporated into the computation of economic benefit. The above problem may be caused by poorly considering the opportunity cost of employment.

Minor comments:
1) What is the difference from an earlier study: Lu, C. The economic benefits and environmental costs of airport operations: Taiwan Taoyuan International Airport, Journal of Air Transport Management, 2011, 17, 360-363? The current paper is very similar to the earlier study. Explain it.
2) “Equation 6” (Line 95, Page 3): This should be “Equation 2.”
3) Equation 2 (Line 98, Page 3): Why no subscript or notation of k is included in the right hand of this equation?
4) “The number of households affected by aircraft noise within different zones is derived by taking the estimated number of households affected for 2012” (Lines 234-235, Page 6): Show the descriptive statistics.
5) “Annual house rent is estimated from the publicly available house price data for the area, considering the average house age and discount rates.” (Lines 236-237, Pages 6-7): The data used for the estimation and the estimation results should be explicitly presented.
6) Tables 4, 5, 7 and Figures 1-3: Show the year of currency? All monetary data is adjusted to 2015 year value?
7) “The average annual income per employee” (Line 283, Page 8): What is the definition of the average annual income? Do they cover all kinds of employees such as part-time workers?
8) “Note that the opportunity cost of the operation of the airport is not considered in this research.” (Lines 288-289, Page 8): Why do you neglect the opportunity cost? From the standard viewpoint of economic evaluation, the operation cost or other factors should be incorporated also. Explain it.
9) Table 7 (Page 9): Show the data of unemployment rate also.
10) Table 7 (Page 9): Why the multiplier is decreasing over year? Explain it.

Author Response

Reviewer 3:
Comments and Suggestions for Authors
General comments:
This paper compares the environmental cost of airport’s operation with its economic benefit,
using a case of Taipei’s airport. Analytical framework is quite straightforward. My opinion is the
estimation of economic benefit is too simple to be biased. The study assumes that all newly
generated employments are hired at full wage rates (or average salary); and it means that the
study assumes such the newly employed labors would be unemployed if the airport would not
be operated. However, the employment generated by the airport operation could be hired in
another industry when the airport would not be operated. In other words, the change of jobs
from less salary to higher salary seems ignored. It is highly afraid that this ignorance leads to
serious overestimation of economic benefit stemming from the airport operation. Both the
change from unemployment to employment and the change from less efficient job to more
efficient job should be explicitly incorporated into the computation of economic benefit. The
above problem may be caused by poorly considering the opportunity cost of employment.
Answer:Thank you for your valuable comments. The author agrees with the views on
the possible over estimate of the economic benefits. Therefore, at the end
of the introduction section, a discussion about opportunity costs and the
defining of the research scope is added to the revised version. In addition,
further research about opportunities about opportunity costs is added in
the conclusion section.
Minor comments:
1) What is the difference from an earlier study: Lu, C. The economic benefits and
environmental costs of airport operations: Taiwan Taoyuan International Airport, Journal of Air
Transport Management, 2011, 17, 360-363? The current paper is very similar to the earlier
study. Explain it.
Answer: The methodologies used in the current paper are similar to the paper
published in 2011. However, this paper has made a big step by exploring
the analysis for 10-year period, applying in a different airport. The purpose
is to investigate the trend in economic benefits and environmental costs,
which would be more convincing than just looking at the data of one year.
In addition, the CO2 emissions from ground operation of the airport have
added to the estimation of the social costs of emissions. All the parameters
used in the empirical analysis have been carefully examined and revised
according to the research to date.
2) “Equation 6” (Line 95, Page 3): This should be “Equation 2.”
Answer: Have revised accordingly.
3) Equation 2 (Line 98, Page 3): Why no subscript or notation of k is included in the right hand
of this equation?
Answer: Have added notation of k in both equations (1) and (2) as well as revising
the texts accordingly.

4) “The number of households affected by aircraft noise within different zones is derived by
taking the estimated number of households affected for 2012” (Lines 234-235, Page 6): Show
the descriptive statistics.
Answer: Have added some explanation regarding number of households in the
revised paper.
5) “Annual house rent is estimated from the publicly available house price data for the area,
considering the average house age and discount rates.” (Lines 236-237, Pages 6-7): The data
used for the estimation and the estimation results should be explicitly presented.
Answer: Have added more explanations about all the data related to the noise
social costs, including the average house age and discount rates. However,
since the house price consists of different years and different administrative
areas, weighted by households, all these data seem to be too trivial to be
presented in the paper. Therefore, the paper shows the most important
parameters of annual house rent and number of households in Table 5.
6) Tables 4, 5, 7 and Figures 1-3: Show the year of currency? All monetary data is adjusted to
2015 year value?
Answer: Yes. Have added the note to these tables.
7) “The average annual income per employee” (Line 283, Page 8): What is the definition of the
average annual income? Do they cover all kinds of employees such as part-time workers?
Answer: It is the average annual income for a full time employee. Have added
“full-time” to the sentence.
8) “Note that the opportunity cost of the operation of the airport is not considered in this
research.” (Lines 288-289, Page 8): Why do you neglect the opportunity cost? From the
standard viewpoint of economic evaluation, the operation cost or other factors should be
incorporated also. Explain it.
Answer:The paper aims to evaluate the gross economic benefits and environmental
costs of an airport. The opportunity costs of employment or airport
operation are not considered in this research. For example, the employment
by the airport could be employed in another industry or even different
regions of Taiwan, or the change of jobs from less salary to higher salary;
whereas the different usage of land might generate different environmental
impacts. All these considerations could lead to various iterative issues. As
the primary purpose of understanding the airport operation on its own, the
scope of the paper is defined to evaluate the airport operation as a closed
loop effect. This is added to the introduction section and discussed again as
future research.
9) Table 7 (Page 9): Show the data of unemployment rate also.

Answer: Since the unemployment rate is not used in the model, therefore, this data
were not collected and presented.
10) Table 7 (Page 9): Why the multiplier is decreasing over year? Explain it.
Answer: As the direct employment is increasing, which will lead to higher indirect
employment, therefore, the multiplier is decreasing over years. This added
to the revised version.

Reviewer 4 Report

The article presented is interesting and above all deals with a very important subject of XXI century.

Besides that the reviewer present some comments to the authors

Literature review:

In academic writing, mainly in the literature review, the authors should mention the names of other authors that write about the same subject in order to justify the importance of the thematic in analysis.

Line 74 - "A number of articles in the literature have dealt with the impacts of exhaust pollutants from 75 different aspects" - WHAT ARTICLES/AUTHORS???? If other researchers want to consult the same articles in order to understand better the subject they can not find the references...

Sometimes a table/figure can resume very good the ideas and the main findings in other articles. In this example the authors could present a table with the impacts of exhaust polluants that and pointed out from other authors.

The authors could enrich the literature review chapter. The reviewer think that is very broad and to simple. If a reader doesn't understand so much about the thematic he/she can not learn here...

Methodology

Is well presented but the reviewer feel that is not connected with the literature review.

The authors could present a table with a resume of other methodologies used by other researchers and explain why the choose this one.

Results

The results could be more explained. Again if the reader don't know so much about the thematic he/she is not going to learn here.

The authors presents several symbols and abbreviations that are not explained in the article.

Conclusions

Very simple. Should be improved.

Author Response

Reviewer 4:
Comments and Suggestions for Authors
The article presented is interesting and above all deals with a very important subject of XXI
century.
Besides that the reviewer present some comments to the authors
Literature review:
In academic writing, mainly in the literature review, the authors should mention the names of
other authors that write about the same subject in order to justify the importance of the
thematic in analysis.
Answer:Thank you for your comments. Yes, it would be clearer. Since the reference
for this paper are mainly from organizations and reports, the author has
chosen this neutral way of writing by not emphasizing particular
organisations.
Line 74 - "A number of articles in the literature have dealt with the impacts of exhaust
pollutants from 75 different aspects" - WHAT ARTICLES/AUTHORS???? If other researchers
want to consult the same articles in order to understand better the subject they can not find the
references...
Answer: The reference has been added to the texts, which also corresponds to
Table 1. In addition, “75” seems to be the number for lines, which is part
of the texts. There might be some formatting problem. Have made sure the
right format in the revised version.
Sometimes a table/figure can resume very good the ideas and the main findings in other
articles. In this example the authors could present a table with the impacts of exhaust polluants
that and pointed out from other authors.
Answer: With regard to their evaluation in monetary terms for emissions, they are
based on the relationship between pollution and damages on human health,
vegetation, buildings, climate change and global warming, with the most
commonly discussed impacts are on human health and climate change.
Since there are some duplications on most of the impacts measured by the
papers, some more sentences have been added to explain the literature
review instead of adding a table.
The authors could enrich the literature review chapter. The reviewer think that is very broad
and to simple. If a reader doesn't understand so much about the thematic he/she can not learn
here...
Answer: There are more discussions added to the introduction section.
Methodology
Is well presented but the reviewer feel that is not connected with the literature review.
The authors could present a table with a resume of other methodologies used by other
researchers and explain why the choose this one.
Answer: Since this paper covers three different topics and methodologies, in order
to make the article concise, the chosen methodology is discussed in texts instead of using tables. The revised version has added more discussion in
the texts.
Results
The results could be more explained. Again if the reader don't know so much about the
thematic he/she is not going to learn here.
The authors presents several symbols and abbreviations that are not explained in the article.
Answer: Have added more discussion in the revised version and checked that all
abbreviations have been explained.
Conclusions
Very simple. Should be improved.
Answer: Have added more discussion in the conclusions as well as
recommendations.

Round 2

Reviewer 2 Report

The authors have done a good job in responding to my comments and the paper now reads quite well. Their explanations are also reasonable and thorough. I am happy with the paper and would recommend it now proceed to publication.

Author Response

Thank you very much.

Reviewer 3 Report

Describe explicitly the differences of this study from the earlier study: Lu, C. The economic benefits and environmental costs of airport operations: Taiwan Taoyuan International Airport, Journal of Air Transport Management, 2011, 17, 360-363.

Author Response

Describe explicitly the differences of this study from the earlier study: Lu, C. The economic benefits and environmental costs of airport operations: Taiwan Taoyuan International Airport, Journal of Air Transport Management, 2011, 17, 360-363.

Answer:

Thank you for your comment. The author has commented on the difference and improvements between this study and Lu’s 2011 paper at the end of section 1. The text is as follows:

The methodologies used in the current paper are similar to those of the paper published in 2011 by Lu [9], which evaluated the employment benefits as well as aircraft noise and emissions impacts of an airport for one year. However, the present paper has advanced on the former work by exploring the analysis over a 10-year period, and applying this to a different airport. The purpose is to investigate the trends in economic benefits and environmental costs, and which circumstances might cause the environmental costs to outweigh the economic benefits that an airport brings. In addition, the CO2 emissions from the ground operation at the airport have added to the estimation of the emission social costs. All the parameters used in the empirical analysis have been carefully examined and revised according to research to date.

Reviewer 4 Report

No comments to add

Author Response

Thank you very much. Have checked through the English.